# Dynamics of stunting from childhood to youthhood in Ethiopia: Evidence from the Young Lives panel data

**Ayalew Astatkie** *

School of Public Health, College of Medicine and Health Sciences, Hawassa University, Hawassa, Ethiopia

* ayalewastatkie@gmail.com

## Abstract

### Introduction

Stunting continues to be a public health challenge with grave health, cognitive and economic consequences. Yet, its dynamics along the life course remain not well investigated in Ethiopia and beyond.

### Methods

Longitudinal data generated by following two (younger and older) cohorts of about 3000 children for nearly 15 years were analyzed to investigate the longitudinal dynamics of stunting in Ethiopia. The cross-sectional prevalence of stunting in each round, longitudinal prevalence, and transition probabilities were determined. Multilevel mixed effects ordinal regression was applied to identify the determinants of stunting accounting for child-level and cluster-level variations.

### Results

The cross-sectional prevalence of severe stunting for the younger cohort fluctuated between 21% and 6%, while for the older cohort it fluctuated between 12% and 3%. Moderate stunting fluctuated between 23% and 16% for the younger cohort and between 22% and 8% for the older cohort. The longitudinal prevalence of severe stunting was 10% in both the younger and older cohorts, whereas that of moderate stunting was 20% for the younger cohort and 18% for the older cohort. Children not stunted at baseline had very high probabilities of remaining not stunted through youthhood (87% for the younger and 90% for the older cohorts). Conversely, children with moderate stunting at baseline had high probabilities either remaining moderately stunted or progressing to severe stunting. Furthermore, children who had severe stunting at baseline had high probabilities of either remaining severely stunted or transitioning to moderate stunting. In both cohorts, older age of the child, female sex, having an educated mother, and being from a household with educated head significantly reduced the risk of stunting. Children from households in the top wealth tertile had a significantly lower risk of stunting in the younger cohort, but not in the older cohort. Similarly,

**Data Availability Statement:** The data underlying the results presented in the study are available from the UK Data Service at http://doi.org/10.5255/UKDA-SN-7483-3.

**Funding:** The author(s) received no specific funding for this work.

**Competing interests:** The author has declared that no competing interests exist.

Productive Safety Net Programme reduced the risk of stunting in the younger cohort, but not in the older cohort.

## Conclusion

Children not stunted early in life are highly likely to grow into non-stunted adults while children stunted early in life are highly likely to grow into stunted adults. Several child-level, maternal, household and programmatic factors affect the risk of stunting. Efforts to prevent stunting shall commence early in life.

## Introduction

Stunting, a state of having a stature too short for one's age, is a devastating nutritional deficiency resulting from poor nutrition in-utero and during early childhood, and due to repeated bouts of infection [1, 2]. Stunting increases the risks of morbidity and mortality among children [1], impairs mental development, and reduces school performance and overall intellectual capacity. Stunted children grow to be economically less productive adults. The effects of stunting transcend generations [1, 2].

Globally, the prevalence of stunting is declining at a slower pace–from 32.5% in 2000 to 21.9% in 2018 [2]. The same trend exists in Ethiopia–stunting declined from 51% in 2005 to 37% in 2019 [3]. Ethiopia remains to be one of the countries where the prevalence of stunting is "very high" [2].

Stunting varies greatly across regions in Ethiopia from 14% in Addis Ababa to 49% in Tigray[3]. Evidence from India shows that such geographical variations in the burden of stunting are explained by multisectoral factors such as gender, education, economic, health, hygiene and demographic factors[4, 5]. The prevalence of stunting in Ethiopia is higher among rural children (41%) than among urban children (26%) [3]. The risk increases with increasing age [3, 6–10] and males are at a higher risk than females [3, 6, 7, 9–11]. Children from poorer households [3, 6, 12–17] and uneducated mothers [3, 6, 7, 13] are at increased risk for stunting. In one Ethiopian study, children with a history of malaria infection have been shown to be at a higher risk of stunting[18]. Previous studies conducted to investigate the relationship between stunting and intestinal parasites did not find statistically significant associations[19–22], except one study which reported a significant association between sever stunting and helminthic infections among girls[23].

While several studies investigating the epidemiology of stunting have been conducted in Ethiopia, most of them focus on under-five children and rely on cross-sectional designs. Consequently, there is considerable lack of evidence regarding the dynamics of stunting across the life course in Ethiopia and beyond. A previous work[24] has investigated the growth pattern, incidence of and recovery from stunting in four countries including Ethiopia, but it was limited to children 1–8 years old. The dynamics of stunting in later life (through youthhood) remain not well understood. Besides, the effect on stunting of social protection programmes such as the Productive Safety Net Programme (PSNP) has not been well investigated.

Therefore, this study investigates the longitudinal dynamics (cross-sectional prevalence, longitudinal prevalence, transition probabilities and determinants) of stunting from childhood to youthhood using the Young Lives data which was generated by following two cohorts of children (younger and older cohorts) for about 15 years in five rounds. The effect on stunting

of participation in PSNP which was introduced in Ethiopia while the Young Lives study was underway has also been investigated in this study.

## Materials and methods

### Study design and population

This study is based on the Young Lives study, which is a longitudinal panel study of approximately 12,000 children carried out over 15 years (in five rounds) in four low- and middle-income countries namely Ethiopia, Peru, Vietnam and India [25, 26]. The present study utilizes the constructed Young Lives dataset for Ethiopia which covers Rounds 1 to 5 [27]. The longitudinal study comprised of two cohorts of children–a younger cohort of 1999 children who were about 1 year old and an older cohort of 1000 children who were about eight years old at the start of the study in 2002 (Round 1). Both cohorts were followed for about 15 years in five rounds–Round 1 (2002), Round 2 (2006), Round 3 (2009), Round 4 (2013) and Round 5 (2016). The younger cohort was followed from 1 year to 15 years of age, while the older cohort was followed from 8 years to 22 years of age. In each round, the study participants in the younger cohort were 1, 5, 8, 12, and 15 years old, respectively while those in the older cohort were 8, 12, 15, 19 and 22 years old, respectively [25, 26].

### Sample size and sampling

The Young Lives sample for Ethiopia, as well as for the other countries, was not intended to be nationally representative, but to be a sample suitable to investigate the longitudinal dynamics of child-related variables and the impact of children's early-life circumstances on children's later outcomes. The sample size was, therefore, decided to be large enough for general statistical analyses such as modeling child welfare and its dynamics overtime. Accordingly, the younger cohort comprised of 1999 study participants and the older cohort comprised of 1000 study participants. Though not incepted to be nationally representative, the sample has been shown to cover children characteristically as diverse as those involved in nationally representative samples such as the Demographic and Health Survey (DHS) and the Welfare Monitoring Survey (WMS) [25, 26, 28]. The number of children who actually participated in each round and the size of the longitudinal attrition are given in Fig 1.

Detailed descriptions of the sampling procedure are given elsewhere[25, 26]. Briefly, the sampling was accomplished using a multistage sampling technique at the start of the study in 2002. In the first stage, out of the nine administrative regions and two city administrations in Ethiopia, four regions–namely Amhara, Oromia, Southern Nations, Nationalities and Peoples (SNNP), and Tigray–and one city administration–namely Addis Ababa–were selected purposefully to ensure national coverage. These five administrative areas account for about 96% of the national population. In the second stage, three to five *woredas* (districts) were selected per region ensuring representation of different poverty levels, urban and rural areas and food deficit status. Totally 20 *woredas* were selected. In the third stage of selection, *kebeles* (lowest administrative units) were selected. At least one *kebele* was selected from each *woreda*. A *kebele* was considered a sentinel site for the panel data collection or was merged with adjacent *kebeles* to form a sentinel site depending on the number of eligible households in each *kebele*. Finally, 100 households with a 1-year old child and 50 households with an 8-year old child were selected randomly from each sentinel site. If a selected household had both a 1-year old child and an 8-year old child, the 1-year old child was selected as the study required larger number of younger children. Poor children were purposively over-sampled.

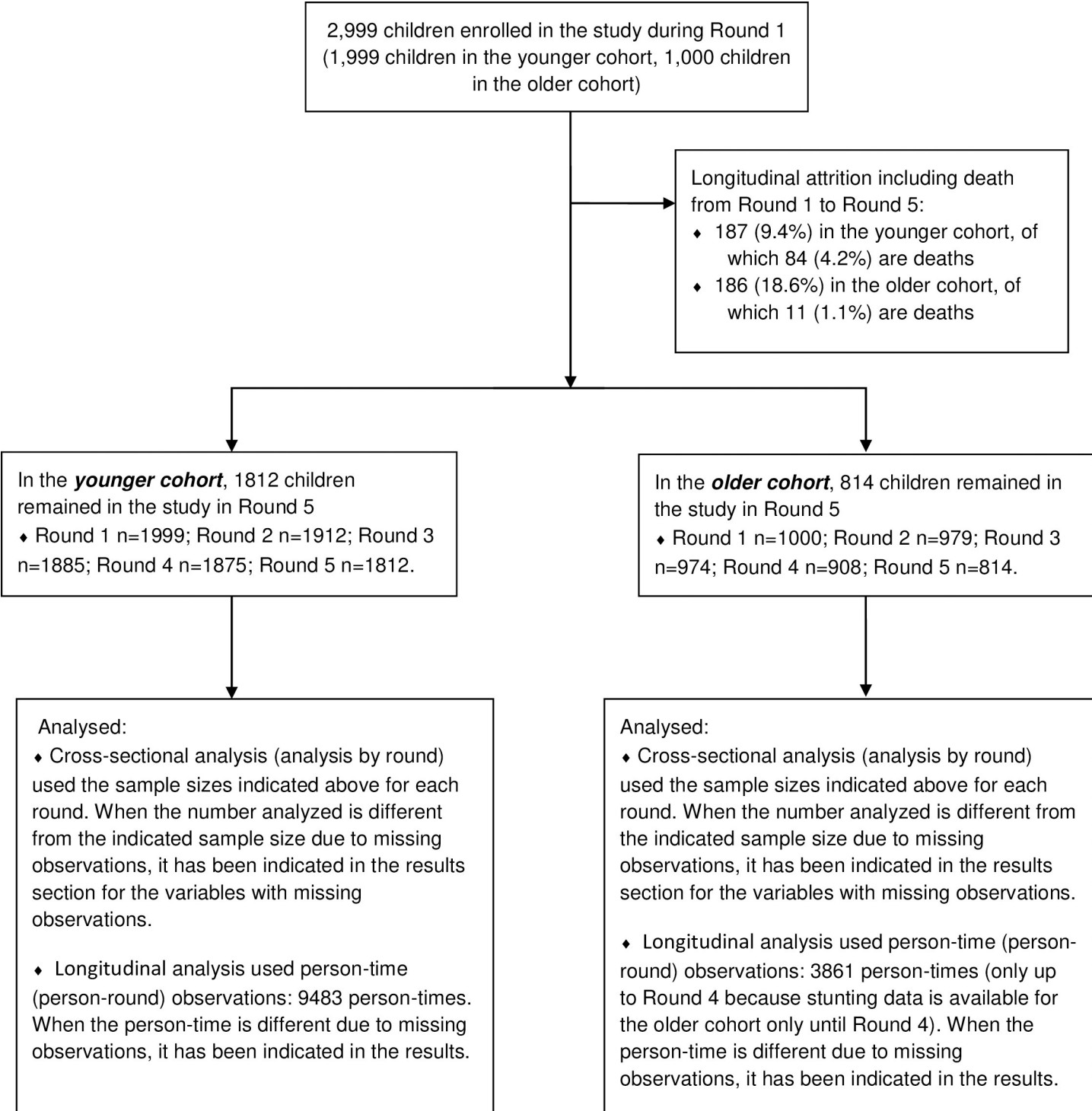

**Fig 1. Flow diagram showing the cohort profile, Ethiopia, 2002–2016.**

## Variables of the study

In the present analysis, the dependent (outcome) variable is stunting. It was measured as an ordinal categorical variable with three mutually exclusive categories, namely *not stunted* (a height-for-age [HFA] z-score of greater than or equal to -2), *moderately stunted* (a HFA of between -3 and -2), and *severely stunted* (a HFA of less than -3) [29]. Three measures of outcome are used in this article–viz., cross-sectional prevalence, longitudinal prevalence and transition probabilities. The cross-sectional prevalence measured the point prevalence of stunting in each round. It is used to show fluctuations in the prevalence of stunting across rounds. The longitudinal prevalence measures the proportion of times a person has the disease (in this case, stunting) in longitudinal studies[30, 31]. It is a useful measure of disease occurrence in longitudinal studies as it avoids problem of defining an outcome in the presence of repeated episodes [30]. Transition probabilities refer to the probability of transitioning (change over time) of categorical variable (in this case, stunting) from one category (level) to another[32].

The Independent variables investigated as possible determinants of stunting included child's age, mother's age, and age of the household head (all in years); child's sex; area of residence (rural vs. urban); levels of education of the mother and of the household head (illiterate, grade 1–4, grade 5–8, above grade 8, other [adult literacy, religious or other]); household size (5 or less vs. greater than 5); household wealth tertile (bottom, middle, top); and participation of at least one household member in PSNP–public works or direct support programme (no vs. yes). The Young Lives wealth index is used as a measure of the socioeconomic status of households[29] and is computed based on three sub-indices, namely housing quality, access to services, and ownership of consumer durables[33]. The PSNP was introduced in Ethiopia to support food insecure households in 2005 (after the Young Lives panel study was launched) [34]. Hence, data on participation in PSNP was collected as of Round 3. For rounds 1 and 2, all households were considered as having not participated in PSNP.

## Methods of data collection

A detailed description of the data collection methodology of the Young Lives study is provided elsewhere [25]. Briefly, the data on which this article is based were collected in each round using interviewer-administered questionnaires from the children (8 years and older) and their primary caregivers. While the core content of the questionnaire remains unchanged across rounds, modifications have been done to take account of life course and contextual changes and based on lessons learned from preceding rounds. Anthropometric measurements such as height have also been taken based on which z-scores were computed to define children's malnutrition status[29].

Data were collected using paper-based questionnaires in the first three rounds. Computer-assisted personal interviewing (CAPI) was implemented in Rounds 4 and 5. Data collectors and supervisors comprised of men and women recruited based on minimum educational requirements and prior experience in data collection. They were all fluent in speaking and writing the languages of the localities in which they were assigned for field work[25]. In all rounds, data collection took place between October and December[28].

## Statistical analysis

The data were analyzed using Stata/IC 15.1 [StataCorp LLC, College Station, Texas, USA]. The data for the younger and older cohort were analyzed separately. Descriptive analyses were performed to obtain summary measures for the basic background characteristics of the study participants and the prevalence and transition probabilities of stunting. The Stata command *xttrans* was used to estimate the transition probabilities of stunting status across rounds.

A multilevel mixed-effects ordered logistic regression with three levels–in which observations in each round were the first-level units, children were the second-level units and clusters (sentinel sites) were the third-level units–was conducted to identify the determinants of stunting accounting for child-level and cluster-level variations. The mixed-effects ordered logistic regression commenced with a crude analysis, in which each potential determinant was examined separately for its possible effect on stunting. Consequently, potential determinants with p-values less than 0.25 on crude analysis were included in the adjusted (multivariable) model. Variables with a large number of missing observations (father's age and father's level of education) were excluded from the adjusted analysis. In addition to missing observations, father's level of education was also found to be redundant with the education level of the household head as evidenced by a high correlation coefficient. Furthermore, caregiver's level of education was excluded from the adjusted model because it correlated highly with mother's level of education. The initial (full) model was successively refined and re-fit by iteratively excluding variables the exclusion of which does not significantly affect the model as a whole (based on likelihood ratio test) and the variables remaining in the model (based on changes in the odds ratios of individual variables).

The importance of the multilevel model over the standard ordinal regression model was tested using likelihood ratio test. Adjusted odds ratios (AORs) with 95% confidence intervals (CIs) were used to determine the presence and strength of association between stunting and potential determinants. AORs with 95% CIs that do not embrace unity (1) were considered statistically significant.

### Ethical considerations

Details of the ethical considerations of the Young Lives study have been described elsewhere [25]. Briefly, the study was conducted in compliance with the ethical standards of the countries in which the study was conducted. The study proposal was reviewed by the London School of Hygiene and Tropical Medicine and the pilot phase of the study was approved by the Rand Afrikaans University in South Africa. Subsequent approvals of the study have been obtained from ethics committees in each of the countries where the Young Lives study was conducted. In Ethiopia, the study was approved by the Institutional Review Board at the College of Health Sciences of Addis Ababa University[28]. Informed consent was obtained in each round from the parents or caregivers and from the children themselves from as early age as possible. The confidentiality and identities of the study participants were protected by excluding names of persons and places from the datasets. Anonymized dataset for this work was obtained from the UK Data Service after submitting the intent of the work (Project id: 118166).

## Results

### Background characteristics of the children

The background characteristics of the children involved in the study, of their parents and the households from which the children were selected are presented in Table 1. For the younger cohort of children, the mean (± standard deviation [SD]) age in Round 1 (2002) was 0.97 (± 0.30) years, whereas in Round 5 (2016) the mean (±SD) age was 15.08 (±0.31) years. The male-to-female ratio across all rounds for the younger cohort was about 1.1. For the older cohort, the mean (±SD) age of the children in Round 1 was 7.88 (±0.29) years, whereas it was 22.03 (±0.31) years in Round 5. The male-to-female ratio for the older cohort was 1.04 in Round 1 and 1.16 in Round 5.

The sample was comprised mostly of rural children for both the younger and older cohorts. The wealth tertile of the households to which the children belonged has improved over the 15

**Table 1. Background characteristics of the study participants, Ethiopia, 2002–2016.**

| Characteristic* | Younger cohort | | | | | Older cohort | | | | |
|---|---|---|---|---|---|---|---|---|---|---|
| | Round 1 | Round 2 | Round 3 | Round 4 | Round 5 | Round 1 | Round 2 | Round 3 | Round 4 | Round 5 |
| **Age of the child (in years), mean (SD)** | 0.97 (0.30) | 5.15 (0.32) | 8.12 (0.34) | 12.12 (0.32) | 15.08 (0.31) | 7.88 (0.29) | 12.05 (0.32) | 15.03 (0.30) | 19.06 (0.33) | 22.03 (0.31) |
| **Age of the mother (in years), mean (SD)** | 27.43 (6.36) | 31.47 (6.37) | 34.37 (6.30) | 38.34 (6.32) | 41.40 (6.31) | 34.08 (7.11) | 38.05 (6.93) | 41.01 (7.06) | 44.95 (6.99) | 47.93 (7.04) |
| **Age of the father (in years), mean (SD)** | 36.64 (9.11) | 40.59 (9.19) | 43.50 (9.10) | 47.23 (8.91) | 50.27 (8.87) | 43.67 (9.52) | 47.76 (9.55) | 50.63 (9.51) | 54.39 (9.28) | 57.34 (9.41) |
| **Sex of child, n (%)** | | | | | | | | | | |
| Male | 1049 (52.48) | 1009 (52.77) | 994 (52.73) | 990 (52.80) | 960 (52.98) | 510 (51.00) | 499 (50.97) | 497 (51.03) | 488 (53.74) | 427 (52.46) |
| Female | 950 (47.52) | 903 (47.23) | 891 (47.27) | 885 (47.20) | 852 (47.02) | 490 (49.00) | 480 (49.03) | 477 (48.97) | 420 (46.26) | 387 (47.54) |
| Total | 1999 (100) | 1912 (100) | 1885 (100) | 1875 (100) | 1812 (100) | 1000 (100) | 979 (100) | 974 (100) | 908 (100) | 814 (100) |
| **Area of residence, n (%)** | | | | | | | | | | |
| Urban | 700 (35.02) | 671 (35.09) | 663 (35.17) | 687 (36.66) | 658 (36.45) | 351 (35.10) | 347 (35.44) | 356 (36.55) | 389 (42.84) | 381 (47.10) |
| Rural | 1299 (64.98) | 1241 (64.91) | 1222 (64.83) | 1187 (63.34) | 1147 (63.55) | 649 (64.9) | 632 (64.56) | 618 (63.45) | 519 (57.16) | 428 (52.90) |
| Total | 1999 (100) | 1912 (100) | 1885 (100) | 1874 (100) | 1805 (100) | 1000 (100) | 979 (100) | 974 (100) | 908 (100) | 809 (100) |
| **Mother's education level, n (%)** | | | | | | | | | | |
| Illiterate | 1180 (59.90) | 929 (50.16) | 807 (44.93) | 680 (38.46) | 652 (38.35) | 556 (60.77) | 433 (49.32) | 417 (49.17) | 297 (37.69) | 261 (37.61) |
| Grade 1 to 4 | 290 (14.72) | 329 (17.28) | 346 (19.27) | 364 (20.59) | 318 (18.71) | 154 (16.83) | 169 (19.25) | 166 (19.58) | 163 (20.69) | 141 (20.32) |
| Grade 5 to 8 | 317 (16.09) | 298 (16.09) | 285 (15.87) | 279 (15.78) | 274 (16.12) | 108 (11.80) | 110 (12.53) | 106 (12.50) | 115 (14.59) | 104 (14.99) |
| Above grade 8 | 162 (8.22) | 183 (9.88) | 184 (10.24) | 202 (11.43) | 204 (12.00) | 61 (6.67) | 63 (7.18) | 59 (6.96) | 64 (8.12) | 53 (7.64) |
| Other (adult literacy, religious or other) | 21 (1.07) | 122 (6.59) | 174 (9.69) | 243 (13.74) | 252 (14.82) | 36 (3.93) | 103 (11.73) | 100 (11.79) | 149 (18.91) | 135 (19.45) |
| Total | 1970 (100) | 1852 (100) | 1796 (100) | 1768 (100) | 1700 (100) | 915 (100) | 878 (100) | 848 (100) | 788 (100) | 694 (100) |
| **Father's education level, n (%)** | | | | | | | | | | |
| Illiterate | 898 (52.27) | 608 (37.55) | 571 (36.77) | 250 (16.49) | 232 (16.07) | 321 (43.73) | 164 (23.40) | 147 (21.91) | 91 (15.17) | 79 (14.91) |
| Grade 1 to 4 | 296 (17.23) | 343 (21.19) | 332 (21.38) | 381 (25.13) | 306 (21.19) | 143 (19.48) | 148 (21.11) | 147 (21.91) | 129 (21.50) | 107 (20.19) |
| Grade 5 to 8 | 262 (15.25) | 283 (17.48) | 270 (17.39) | 329 (21.70) | 312 (21.61) | 127 (17.30) | 146 (20.83) | 142 (21.16) | 130 (21.67) | 119 (22.45) |
| Above grade 8 | 239 (13.91) | 246 (15.19) | 228 (14.68) | 270 (17.81) | 273 (18.91) | 93 (12.67) | 92 (13.12) | 89 (13.26) | 86 (14.33) | 70 (13.21) |
| Other (adult literacy, religious or other) | 23 (1.34) | 139 (8.59) | 152 (9.79) | 286 (18.87) | 321 (22.23) | 50 (6.81) | 151 (21.54) | 146 (21.76) | 164 (27.33) | 155 (29.25) |
| Total | 1718 (100) | 1619 (100) | 1553 (100) | 1516 (100) | 1444 (100) | 734 (100) | 701 (100) | 671 (100) | 600 (100) | 530 (100) |
| **Region of residence, n (%)** | | | | | | | | | | |
| Tigray | 400 (20.01) | 385 (20.14) | 383 (20.32) | 382 (20.42) | 368 (20.39) | 201 (20.10) | 201 (20.53) | 200 (20.53) | 186 (20.53) | 151 (18.69) |
| Amhara | 400 (20.01) | 383 (20.03) | 381 (20.21) | 367 (19.62) | 355 (19.67) | 200 (20.00) | 192 (19.61) | 192 (19.71) | 176 (19.43) | 160 (19.80) |
| Oromiya | 399 (19.96) | 385 (20.14) | 383 (20.32) | 392 (20.95) | 380 (20.15) | 199 (19.90) | 199 (20.33) | 201 (20.64) | 199 (21.96) | 172 (21.29) |
| SNNP | 500 (25.01) | 479 (25.05) | 472 (25.04) | 463 (24.75) | 452 (25.04) | 250 (25.00) | 244 (24.92) | 235 (24.13) | 198 (21.85) | 192 (23.76) |

(*Continued*)

**Table 1.** (Continued)

| Characteristic* | Younger cohort | | | | | Older cohort | | | | |
|---|---|---|---|---|---|---|---|---|---|---|
| | Round 1 | Round 2 | Round 3 | Round 4 | Round 5 | Round 1 | Round 2 | Round 3 | Round 4 | Round 5 |
| Addis Ababa City Administration | 300 (15.01) | 280 (14.64) | 266 (14.11) | 267 (14.27) | 250 (13.85) | 150 (15.00) | 143 (14.61) | 146 (14.99) | 147 (16.23) | 133 (16.46) |
| Total | 1999 (100) | 1912 (100) | 1885 (100) | 1871 (100) | 1805 (100) | 1,000 (100) | 979 (100) | 974 (100) | 906 (100) | 808 (100) |
| **Wealth tertile, n (%)** | | | | | | | | | | |
| Bottom | 1185 (59.94) | 836 (43.95) | 606 (32.17) | 430 (22.98) | 265 (14.69) | 572 (57.37) | 386 (39.47) | 248 (25.54) | 120 (13.32) | 63 (7.79) |
| Middle | 452 (22.86) | 549 (28.86) | 661 (35.08) | 699 (37.36) | 750 (41.57) | 270 (27.08) | 301 (30.78) | 380 (39.13) | 330 (36.63) | 311 (38.44) |
| Top | 340 (17.20) | 517 (27.18) | 617 (32.75) | 742 (39.66) | 789 (43.74) | 155 (15.55) | 291 (29.75) | 343 (35.32) | 451 (50.06) | 435 (53.77) |
| Total | 1977 (100) | 1902 (100) | 1884 (100) | 1871 (100) | 1804 (100) | 997 (100) | 978 (100) | 971 (100) | 901 (100) | 809 (100) |
| **Household size, mean (SD)** | 5.72 (2.16) | 6.05 (2.08) | 6.19 (1.98) | 5.88 (1.93) | 5.77 (1.93) | 6.44 (2.16) | 6.50 (2.05) | 6.35 (2.12) | 5.37 (2.29) | 4.63 (2.28) |

* Sample sizes differ from variable to variable and across rounds due to missing observations and longitudinal attrition.

Note: *n*, *number*; *SD*, *standard deviation; SNNP, Southern Nations, Nationalities and Peoples*

years follow-up time (Round 1 to Round 5). For the younger cohort, about 60% of the children were from the bottom wealth tertile households in Round 1 but only 15% were from the bottom wealth tertile households in Round 5. Similarly, for the older cohort 57% of the children belonged to households from the bottom wealth tertile in Round 1, while only 8% belonged to households of the bottom wealth tertile in Round 5.

Among households from which the younger cohort of children were recruited, 28% participated (by at least one household member) in PSNP in the 12 months time preceding the interview in Round 3, 21% participated in Round 4 and 16% participated in Round 5. Among households from which the older cohort of children were selected, 28% participated in PSNP in Round 3, 18% participated in Round 4 and 11% participated in Round 5.

## Cross-sectional and longitudinal prevalence of stunting across the life course

The cross-sectional (point) prevalence of stunting fluctuated across the five rounds for both the younger and older cohorts, but showed an overall decremental trend. For the younger cohort, the highest cross-sectional prevalence of severe stunting (20.87%) was observed in Round 1 (2002, at 1 year) and of moderate stunting (23.17%) was observed in Round 2 (2006, at 5 years). The lowest cross-sectional prevalence of both severe stunting (5.7%) and moderate stunting (15.77%) for the younger cohort was observed in Round 3 (2009, at 8 years) (Fig 2). For the older cohort, the highest cross-sectional prevalence of severe stunting (11.93%) was recorded in Round 1 (2002, at 8 years) and that of moderate stunting (21.66%) was observed in Round 2 (2006, at 12 years). The lowest cross-sectional prevalence of both severe stunting (2.94%) and moderate stunting (7.76%) for the older cohort were recorded in Round 4 (2013, at 19 years) (Fig 3) (height-for-age was not computed for the older cohort in Round 5 as the World Health Organization [WHO] reference tables do not apply at that age; hence, no stunting data for the older cohort in Round 5 [29]). For both cohorts, the cross-sectional prevalence of moderate stunting remained higher than that of severe stunting across all rounds.

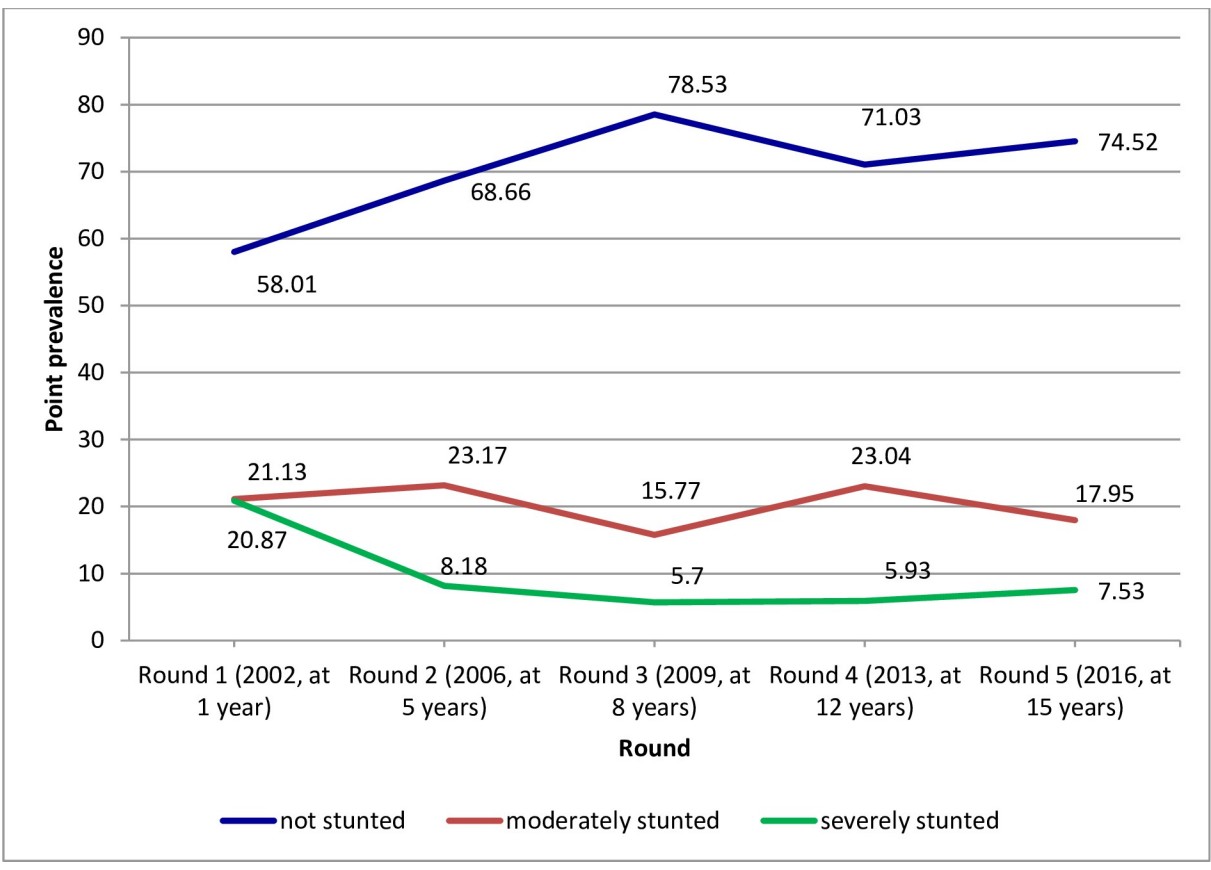

**Fig 2. Cross-sectional (point) prevalence of stunting in the younger cohort across the five rounds (childhood to youthhood), Ethiopia, 2002–2016.**

In the five rounds of follow up, the proportion of times the younger cohort had moderate stunting (the longitudinal prevalence of moderate stunting) was 20.24% (95% CI: 19.44% - 21.06%), whereas in the same cohort the longitudinal prevalence of severe stunting was 9.70% (95% CI: 9.12% - 10.32%). For the older cohort, the longitudinal prevalence of moderate stunting across the four rounds of follow up (Rounds 1–4) was 18.13% (95% CI: 16.86% - 19.46%) while that of severe stunting was 9.54% (95% CI: 8.59% - 10.58) (Table 2).

### Transition probabilities of stunting status over time

In the younger cohort, children who had no stunting at baseline (at 1 year of age) had 87% probability of remaining not stunted in each round whereas they had 11% probability of transitioning into moderate stunting and 2% probability of transitioning into severe stunting across the life course to youthhood. On the other hand, younger children who had moderate stunting at baseline (at 1 year of age) had 40% probability of remaining moderately stunted in each round while they had 47% probability of transitioning into no stunting (recovery) and 13% probability of transitioning into severe stunting. Further, younger children who initially (at 1 year of age) had severe stunting had 28% probability of remaining in severe stunting in each round while they had 32% probability of transitioning into no stunting and 40% probability of transitioning into moderate stunting.

In the older cohort, children who initially (at 8 years of age) had no stunting had 90% probability of remaining not stunted in each round much as they had 9% probability of

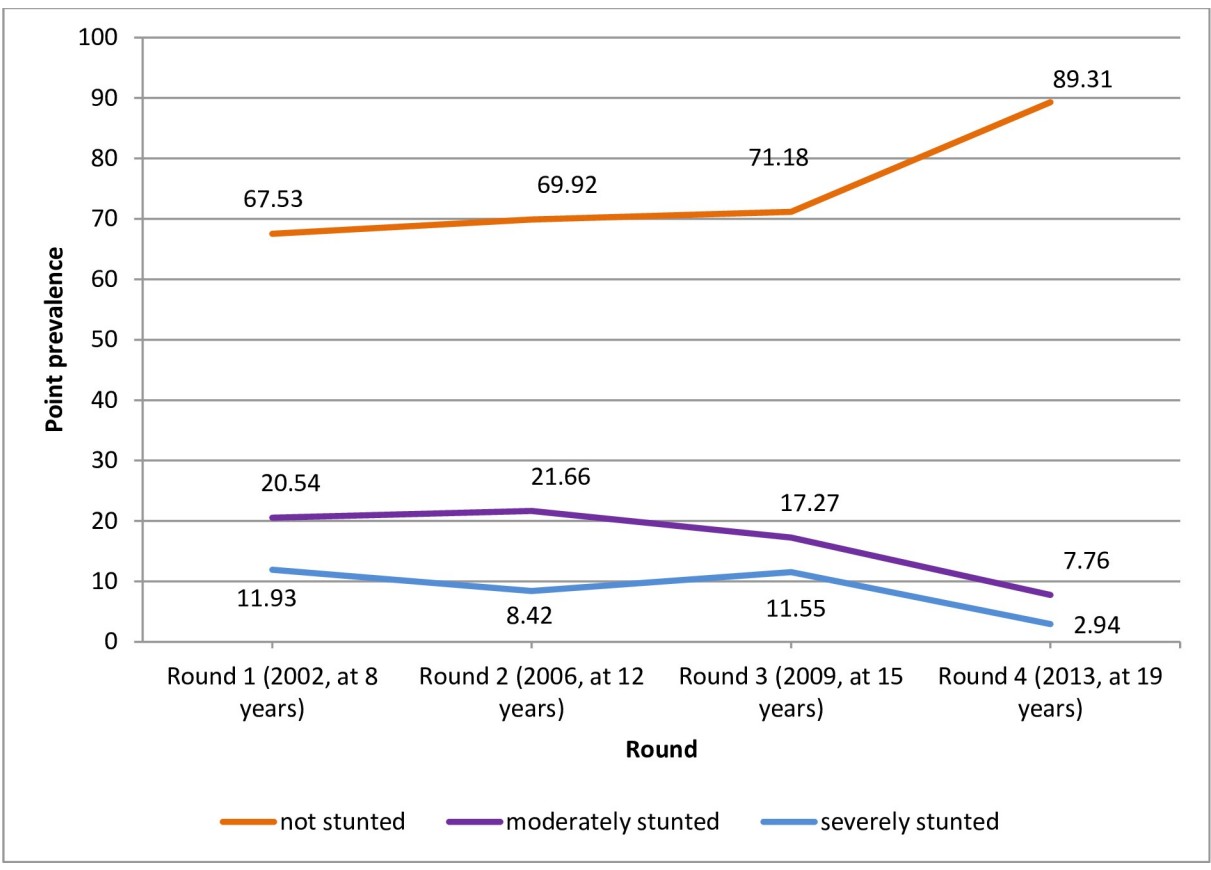

**Fig 3. Cross-sectional (point) prevalence of stunting in the older cohort across the four rounds (childhood to youthhood), Ethiopia, 2002–2013.**

transitioning into moderate stunting and 1% probability of transitioning into severe stunting across the life course to youthhood. Conversely, older children who had moderate stunting at baseline (at 8 years of age) had 36% probability of remaining in moderate stunting whilst they had 51% probability of transitioning into no stunting and 13% probability of transitioning into severe stunting. Besides, older children who initially (at 8 years of age) had severe stunting had 46% probability of remaining in severe stunting while they had 20% probability of transitioning into no stunting and 34% probability of transitioning into moderate stunting (Table 3).

**Table 2. Longitudinal prevalence of stunting from childhood to youthhood, Ethiopia, 2002–2016.**

| Stunting | Number | Prevalence (%) | 95% confidence interval |
|---|---|---|---|
| **Stunting in the younger cohort** (person-time n = 9378) | | | |
| No stunting | 6570 | 70.06 | 69.12% - 70.98% |
| Moderate stunting | 1898 | 20.24 | 19.44% - 21.06% |
| Severe stunting | 910 | 9.70 | 9.12% - 10.32% |
| **Stunting in the older cohort** (person-time n = 3376) | | | |
| No stunting | 2442 | 72.33 | 70.80% - 73.82% |
| Moderate stunting | 612 | 18.13 | 16.86% - 19.46% |
| Severe stunting | 322 | 9.54 | 8.59% - 10.58 |

**Table 3. Transition probabilities of stunting status from childhood to youthhood, Ethiopia, 2002–2016.**

| | | Final stunting status (%) | | |
|---|---|---|---|---|
| | | No stunting | Moderate stunting | Severe stunting |
| Younger cohort (person-time n = 7365) | | | | |
| Initial stunting status | No stunting | 87.27 | 10.99 | 1.74 |
| | Moderate stunting | 46.56 | 40.21 | 13.23 |
| | Severe stunting | 31.54 | 40.08 | 28.37 |
| Older cohort (person-time n = 2368) | | | | |
| Initial stunting status | No stunting | 89.96 | 8.69 | 1.36 |
| | Moderate stunting | 51.21 | 35.63 | 13.16 |
| | Severe stunting | 19.52 | 34.26 | 46.22 |

## Determinants of stunting

In the younger cohort, child's age, child's sex, area of residence, mother's level of education, level of education of the household head, household wealth status and participation of households in PSNP were statistically significant determinants of stunting. With a one year increase in the age of the child, the odds of severe stunting versus the combined categories of moderate stunting and no stunting, and the odds of the combined moderate and severe stunting versus no stunting decreases by 8% (AOR 0.92; 95% CI: 0.90, 0.94). Females had a 44% lesser odds of stunting (AOR: 0.56; 0.46, 0.69), whereas rural children had 59% excess odds of stunting relative to urban children (AOR: 1.59; 95% CI: 1.06, 2.38). Children of educated mothers and children from households with educated heads had lower odds of stunting. Similarly, children from the top wealth tertile households and children from households participating in PSNP had lower odds of stunting (Table 4).

In the older cohort, age of the child, age of the mother, sex of the child, area of residence, level of education of the mother and level of education of the household head were significant determinants of stunting. Increase both in the age of the child and of the mother decreases the odds of stunting. While being a female child decreases the odds of stunting, being a rural child increases the odds of stunting. Similarly, children of educated mothers and children from households with educated heads had lower odds of stunting. Household wealth status and participation in PSNP did not have significant effects on stunting in the older cohort (Table 5).

## Discussion

The present further analysis of the Young Lives data shows that the cross-sectional prevalence of stunting fluctuates from childhood to youthhood, but shows a general, albeit sluggish, decremental trend. The highest cross-sectional prevalence of severe stunting for both the younger (21%) and older (12%) cohorts was observed at baseline (Round 1). The lowest cross-sectional prevalence of severe stunting for the younger cohort (5.7%) was observed in Round 3 and for the older cohort (3%) in Round 4 (i.e., the last round height-for-age was measured for the older cohort or at age 19 years). The highest cross-sectional prevalence of moderate stunting for both the younger cohort (23.2%) and the older cohort (21.7%) was observed in Round 2, whereas the lowest cross-sectional prevalence for the younger cohort (16%) was in Round 3 and for the older cohort (8%) in Round 4.

The highest cross-sectional prevalence of stunting at baseline and Round 2 for both cohorts and the decline, though slight, afterwards could be attributed to falling poverty rates, and improvements overtime in agricultural productivity, food security and nutrition in Ethiopia [35] and implementation of social protection programmes such as the PSNP to improve the

**Table 4. Determinants of stunting in the younger cohort, Ethiopia, 2002–2016.**

| Variable | Total | Stunting, n (%) | | COR (95% CI) | AOR (95% CI)*¶ |
|---|---|---|---|---|---|
| | | **Moderate** | **Severe** | | |
| **Age of the child**, in years (n = 9,477) | 9,477 | *Continuous covariate* | *Continuous covariate* | **0.91 (0.90, 0.92)** | **0.92 (0.90, 0.94)** |
| **Age of the mother**, in years (n = 9041) | 9,041 | *Continuous covariate* | *Continuous covariate* | **0.94 (0.93, 0.95)** | 1.01 (1.00, 1.03) |
| **Age of the household head**, in years (n = 9460) | 9460 | *Continuous covariate* | *Continuous covariate* | **0.97 (0.97, 0.98)** | 0.99 (0.98, 1.00) |
| **Sex of the child** (n = 9,378) | | | | | |
| Male | 4,947 | 1,110 (22.4) | 559 (11.3) | 1 | 1 |
| Female | 4, 431 | 788 (17.8) | 351 (7.9) | **0.57 (0.47, 0.70)** | **0.56 (0.46, 0.69)** |
| **Area of residence** (n = 9,371) | | | | | |
| Urban | 3,348 | 519 (15.5) | 195 (5.8) | 1 | 1 |
| Rural | 6, 023 | 1,379 (22.9) | 714 (11.9) | **2.59 (1.79, 3.74)** | **1.59 (1.06, 2.38)** |
| **Mother's level of education** (n = 8,987) | | | | | |
| Illiterate | 4,194 | 1,000 (23.8) | 545 (13.0) | 1 | 1 |
| Grade 1 to 4 | 1,624 | 300 (18.5) | 133 (8.2) | **0.51 (0.40, 0.64)** | **0.74 (0.57, 0.97)** |
| Grade 5 to 8 | 1,436 | 225 (15.7) | 91 (6.3) | **0.45 (0.34, 0.60)** | **0.69 (0.49, 0.96)** |
| Above grade 8 | 923 | 111 (12.0) | 40 (4.3) | **0.26 (0.18, 0.38)** | **0.58 (0.37, 0.91)** |
| Other (adult literacy, religious education) | 810 | 189 (23.3) | 72 (8.9) | **0.60 (0.46, 0.78)** | 0.97 (0.72, 1.31) |
| **Level of education of the household head** (n = 9,213) | | | | | |
| Illiterate | 3,206 | 742 (23.1) | 438 (13.7) | 1 | 1 |
| Grade 1 to 4 | 1,933 | 389 (20.1) | 172 (8.9) | **0.57 (0.47, 0.69)** | 0.93 (0.74, 1.17) |
| Grade 5 to 8 | 1,616 | 301 (18.6) | 106 (6.6) | **0.48 (0.38, 0.61)** | 0.95 (0.72, 1.25) |
| Above grade 8 | 1,266 | 136 (10.7) | 80 (6.3) | **0.30 (0.22, 0.41)** | **0.69 (0.47, 0.999)** |
| Other (adult literacy, religious education) | 1,192 | 295 (24.8) | 103 (8.6) | **0.63 (0.51, 0.78)** | **1.36 (1.05, 1.76)** |
| **Household wealth tertile** (n = 9,338) | | | | | |
| Bottom tertile | 3,266 | 821 (25.1) | 502 (15.4) | 1 | 1 |
| Middle tertile | 3,086 | 677 (21.9) | 267 (8.7) | **0.52 (0.45, 0.61)** | 0.85 (0.71, 1.01) |
| Top tertile | 2,986 | 396 (13.3) | 134 (4.5) | **0.27 (0.22, 0.33)** | **0.63 (0.48, 0.83)** |
| **Household participation in PSNP** (n = 9,370) | | | | | |
| No | 8,154 | 1,632 (20.0) | 814 (10.0) | 1 | 1 |
| Yes | 1,216 | 266 (21.9) | 94 (7.7) | **0.50 (0.42, 0.60)** | **0.71 (0.58, 0.87)** |

\* Note: Likelihood ratio test comparing the multilevel model vs. the standard ordinal regression model: Chi-square at a degree of freedom of 2 = 1429.21; p < 0.001.

¶ n for the adjusted analysis = 8,765.

Odds ratios in bold type face indicate statistically significant associations.

AOR, adjusted odds ratio; CI, confidence interval; COR, crude odds ratio; n, number of observed events (longitudinal or person-time); PSNP, Productive Safety Net Programme; %, longitudinal prevalence

food security of impoverished households [34]. Fluctuations of the prevalence overtime may imply periodic effects of drought and consequent food shortage in the country. Besides, the highest cross-sectional prevalence of moderate stunting in both cohorts in Round 2 may result, at least partly, from transitioning from severe to moderate stunting as discussed below. Further, the lowest cross-sectional prevalence of both severe and moderate stunting in the older cohort in Round 4 (by age 19) and the general decremental trend of stunting in both the younger and older cohorts imply the occurrence of catch-up growth compensating the growth deficit faced in earlier ages. An earlier analysis of growth patterns for children 1–8 years[24] has also shown significant increases in height-for-age z-scores (HAZ) and decrease in stunting prevalence across rounds corroborating the current evidence. Thus, while preventing

**Table 5. Determinants of stunting in the older cohort, Ethiopia, 2002–2013.**

| Variable | Total | Stunting, n (%) | | COR (95% CI) | AOR (95% CI)* ¶ |
|---|---|---|---|---|---|
| | | Moderate | Severe | | |
| **Age of the child**, in years (n = 3,855) | 3,855 | *Continuous covariate* | *Continuous covariate* | **0.86 (0.84, 0.89)** | **0.92 (0.87, 0.97)** |
| **Age of the mother**, in years (n = 3,398) | 3,398 | *Continuous covariate* | *Continuous covariate* | **0.91(0.89, 0.93)** | **0.96 (0.93, 0.99)** |
| **Sex of the child** (n = 3,376) | | | | | |
| Male | 1,730 | 332 (19.2) | 217 (12.5) | 1 | 1 |
| Female | 1,646 | 280 (17.0) | 105 (6.4) | **0.41 (0.28, 0.61)** | **0.40 (0.26, 0.62)** |
| **Area of residence** (n = 3,376) | | | | | |
| Urban | 1,245 | 164 (13.2) | 65 (5.2) | 1 | 1 |
| Rural | 2,131 | 448 (21.0) | 257 (12.1) | **3.71 (1.82, 7.58)** | 2.38 (0.99, 5.73) |
| **Mother's level of education** (n = 3,010) | | | | | |
| Illiterate | 1,551 | 350 (22.6) | 170 (11.0) | 1 | 1 |
| Grade 1 to 4 | 563 | 95 (16.9) | 71 (12.6) | 1.23 (0.73, 2.06) | 1.48 (0.84, 2.62) |
| Grade 5 to 8 | 374 | 40 (10.7) | 24 (6.4) | 0.55 (0.29, 1.07) | **0.36 (0.16, 0.80)** |
| Above grade 8 | 215 | 27 (12.6) | 12 (5.6) | 0.92 (0.38, 2.19) | 0.59 (0.20, 1.72) |
| Other (adult literacy, religious education) | 307 | 41 (13.4) | 17 (5.5) | **0.32 (0.17, 0.61)** | 0.61 (0.32, 1.16) |
| **Level of education of the household head** (n = 3,276) | | | | | |
| Illiterate | 1,123 | 268 (23.9) | 132 (11.8) | 1 | 1 |
| Grade 1 to 4 | 637 | 105 (16.5) | 54 (8.5) | **0.56 (0.37, 0.86)** | **0.60 (0.36, 0.99)** |
| Grade 5 to 8 | 597 | 82 (13.7) | 54 (9.1) | **0.47 (0.27, 0.79)** | 0.72 (0.38, 1.37) |
| Above grade 8 | 401 | 50 (12.5) | 25 (6.2) | **0.46 (0.24, 0.87)** | 0.94 (0.41, 2.14) |
| Other (adult literacy, religious education) | 518 | 87 (16.8) | 47 (9.1) | **0.41 (0.26, 0.62)** | 0.99 (0.58, 1.67) |
| **Household size** (n = 3,376) | | | | | |
| 5 or less | 1,218 | 205 (16.8) | 97 (8.0) | 1 | 1 |
| > 5 | 2,158 | 407 (18.9) | 225 (10.4) | **1.47 (1.09, 1.98)** | 1.26 (0.89, 1.78) |
| **Household wealth tertile** (n = 3,367) | | | | | |
| Bottom tertile | 1,242 | 308 (24.8) | 181 (14.6) | 1 | 1 |
| Middle tertile | 1,107 | 190 (17.2) | 99 (8.9) | **0.48 (0.36, 0.65)** | 0.78 (0.54, 1.11) |
| Top tertile | 1,018 | 113 (11.1) | 40 (3.9) | **0.20 (0.13, 0.31)** | 0.59 (0.34, 1.05) |
| **Household participation in PSNP** (n = 3,376) | | | | | |
| No | 3,021 | 540 (17.9) | 283 (9.4) | 1 | 1 |
| Yes | 355 | 72 (20.3) | 39 (11.0) | **0.62 (0.44, 0.87)** | 1.30 (0.86, 1.97) |

* Note: Likelihood ratio test comparing the multilevel model vs. the standard ordinal regression model: Chi-square at a degree of freedom of 2 = 616.30; p < 0.001.

¶ n for the adjusted analysis = 2,894 (Round 5 not included as there was no stunting data for the older cohort in Round 5).

Odds ratios in bold type face indicate statistically significant associations.

AOR, adjusted odds ratio; CI, confidence interval; COR, crude odds ratio; n, number of observed events (longitudinal or person-time); PSNP, Productive Safety Net Programme; %, longitudinal prevalence.

undernutrition and the consequent growth deficit since in-utero life is of prime necessity, for children who suffered from linear growth restriction in earlier life, subsequent interventions could result in considerable catch-up growth along the life course to youthhood.

The longitudinal prevalence of severe and moderate stunting for the younger cohort was 10% and 20%, respectively. Similarly, the longitudinal prevalence of severe and moderate stunting for the older cohort was 10% and 18%, respectively. This implies that with repeated measurements of stunting across the life course to youthhood, the mean probability of being stunted in both the younger and older cohort of children is similar. The overall longitudinal prevalence of stunting (severe and moderate combined) for the younger and older cohorts are

30% and 28%, respectively, revealing that along the life course from childhood to youthhood, the longitudinal prevalence of stunting in Ethiopia falls in the range of "high" and "very high" as per the recent United Nations Children's Fund (UNICEF) / WHO / World Bank Group cut-offs for classifying the public health significance of stunting [2].

Among children not stunted at baseline, there is a high probability of remaining not stunted as they grow to youthhood (87% among younger children and 90% among older children). This implies the importance of preventing growth deficits early in life (in-utero and early childhood) to substantially reduce the risk of stunting in later life and curb the transgenerational effect of stunting. Conversely, among younger children moderately stunted at baseline (at 1 year of age), there is 47% probability of transitioning to no stunting across the life course to youthhood, while there is 40% probability of remaining moderately stunted and 13% probability of transitioning to severe stunting. Similarly, among older children moderately stunted at baseline (at 8 years of age), there is 51% probability of transitioning to no stunting overtime, while there is 36% probability of remaining moderately stunted and 13% probability of transitioning to severe stunting. These findings clearly suggest that while Ethiopian children affected by moderate stunting early in life have a considerable (nearly 50%) probability of going through sufficient catch-up growth to attain recovery as they grow, they also are at a higher risk of remaining moderately stunted or progressing to severe stunting.

Furthermore, younger children who are severely stunted at baseline (at 1 year of age) have 40% probability of transitioning to moderate stunting, while they have 32% probability of transitioning to no stunting and 28% probability of remaining severely stunted as they grow to youthhood. On the other hand, older children who are severely stunted at baseline (at 8 years of age) have 46% probability of remaining severely stunted, whilst they have 34% probability of transitioning to moderate stunting and 20% probability of transitioning to no stunting. These findings reveal that children severely stunted early in life have a greater risk of growing in to stunted adults, though a considerable proportion of them may transition to moderate stunting, corroborating the argument that stunted children "may never attain their full possible height"[2]. Therefore, whilst intervention efforts once stunting ensues might play a considerable role in ameliorating the severity or attaining recovery, prevention efforts early in life would better serve the purpose of preventing stunting.

In both the younger and older cohorts, a 1-year increase in the age of the child decreases the odds of stunting by about 8%. This result is in contrast to several previous reports[3, 6–9] which showed an increased risk of stunting with increase in age. However, the previous studies were limited to only children below five years[3, 6, 9] or 6–14 years[7, 8] and used cross-sectional designs, which may account for the observed difference. Conversely, the present result is consistent with previous findings by Lundeen et al[24] and Crookston et al[36] based on analysis of the Young Lives panel data for children 1–8 years. The decrease in the risk of stunting with increase in age could be related to contextual changes (socioeconomic and nutritional improvements) and implementation of social protection programmes and subsequent catch-up growth across the life course of the children as discussed above.

Females have a considerably lower risk of stunting compared to males, which is in agreement with several previous reports[6, 7, 9, 11, 37, 38]. About three decades back, Svedberg[39] has proposed that the relative advantage of females relative to males vis-à-vis nutritional status (and mortality) in Sub-Saharan Africa (SSA) could be due to preferential treatment of girls because of the economic gain they bring through their "more active part in agriculture and food production". In a follow-up study of children from early infancy to three years of age in Senegal, Bork and Diallo[37] have shown that male babies commence complementary feeding at an earlier age (2–3 months) than female babies. As such, they argued that differences in infant feeding practices could account for the increased risk of stunting in male children.

However, based on an analysis of Demographic and Health Surveys and World Fertility Surveys data for SSA, Garenne[40] showed that the same level of care (preventive, curative, feeding practice) is accorded to male and female children. Thus, according to Garenne, gender bias in child care is unlikely to explain the higher risk of stunting among males. On the other hand, Wamani and colleagues[38] have hypothesized that the use of different growth references for males and females by the WHO/National Centre for Health Statistics (NCHS) might contribute to observed differences in stunting between males and females. However, these authors also argue that, sex differences in stunting disappear in the socioeconomically advantaged children. Accordingly, they contend that had the difference in stunting been truly related to difference in the reference for males and females, stunting differential between the sexes would have been the same across varying socioeconomic gradients. Now, an explanation proposed to more likely be the basis of the difference in stunting between males and females is a biological mechanism[37, 38, 40], such as increased susceptibility of male children to infectious diseases[37], but it remains to be a subject of further research[38].

Rural residence significantly increases the risk of stunting in the younger cohort but not in the older cohort. This result is consistent with the finding of the Ethiopian mini-DHS 2019 which showed a higher prevalence of stunting among rural underfive children compared to urban children[3]. Heady et al[41], based on analysis of Demographic and Health Surveys from 23 countries in SSA, have shown that the nutritional disadvantage of rural children vis-à-vis the urban ones is related to disadvantages in wealth, education and infrastructure services in rural areas. Why the effect of urban-rural residence disappears in the older cohort in the present analysis is unclear, though.

Maternal education decreases the risk of stunting in both the younger and older cohorts as does education of the household head. The inverse association between maternal education and stunting has been documented in several previous studies[3, 6, 9, 13, 42]. The possible mechanisms by which maternal education may improve child health and nutrition have been discussed in detail by Günes[43]. According to Günes, educated mothers initiate preventive care earlier, engage less in behaviours that pose health risk to the mother and the baby such as smoking, and have reduced fertility and delayed first birth. Similar mechanisms have also been forwarded by Currie and Moretti[44]. As such, children of educated mothers are likely to have better nutritional status and overall health condition. On the other hand, based on analysis of DHS data for 22 developing countries, Desai and Alva[45] argue that maternal education is "a proxy for the socioeconomic status of the family and geographic area of residence". Accordingly, improved child health and nutrition may not be the effects of maternal education per se, but a result of better off socioeconomic and geographic contexts which educated mothers are likely to be in. However, Desai and Alva also show that "children of educated mothers are more likely to engage in health-promoting behaviour", and hence are likely to have health and nutritional advantages. So, while the mechanism by which maternal education impacts the nutrition and health of children remains to be a subject of further research, the positive correlation between maternal education and child health and nutrition is clearly visible. The effect of literacy (educational status) of the household head may also follow similar mechanisms as that of the maternal education.

In the younger cohort, children from the top wealth tertile households have lower risk of stunting, but this was not the case for the older cohort. The reduced risk of stunting in the upper wealth category vis-à-vis the lowest wealth category, which is consistent with previous findings[3, 6, 12, 13, 46], implies the presence of socioeconomic inequality in the risk of stunting across the life course of children in Ethiopia. The higher risk to stunting of socioeconomically disadvantaged children seems to be a consequence of insufficient food intake, increased susceptibility to diseases and exposure to health risks, and lack of access to preventive and

curative health services[46–48]. Why the association between wealth and stunting disappears in the older cohort is unclear.

Participation of household members in PSNP significantly reduces the risk of stunting in the younger cohort, but not in the older cohort. The PSNP, launched in 2005, aims to provide social protection to food insecure households such that such households get food secure and fulfill basic necessities without depleting household assets[34, 49]. It mostly operates in two modalities–Public Works Programme offered to poor, food insecure households that have at least one "able-bodied labour power", and a Direct Support Programme that targets households poorer than those covered by the Public Works Programme and lack labour power[49]. Beneficiaries of the Public Works Programme receive payments either in cash or in type (cereals) in exchange for labour. Those in the Direct Support Programme receive support but are not required to provide labour force. Households graduate from the programme when evaluated as having built sufficient asset [34, 49].

In the present analysis, participation in PSNP reduced the odds of stunting by 29% in the younger cohort. A previous analysis of the first three rounds of the Young Lives data by Porter and Goyal[50] has also shown that children from households participating in PSNP are nutritionally advantaged based on HAZ and weight-for-age z-score (WAZ) measurements. According to Berhane et al[51], PSNP significantly improves food security and increases the meal children consume. In a different work, Berhane et al[49] show that PSNP reduces child labour load, especially as PSNP payments improve overtime. Hence, improvements in household food security coupled with reduction in child labour might explain how PSNP participation impacts children's nutritional status, as also argued by Porter and Goyal[50].

However, the result of the present analysis regarding the impact of PSNP on stunting stands in contrast to the findings of Berhane et al[49] which show absence of a significant effect of PSNP on stunting. Yet, there are methodological differences between Berhane et al's study and the present work, which may in some way account for the differences. First, Berhane et al's study of the nutritional impact of PSNP was limited to only underfive children and follow-up was for a shorter period of time (about 4 years). Hence, nutritional improvements that appear later in life might be missed. Second, Berhane et al did a cross-sectional analysis separately for each round of their longitudinal data, and hence the longitudinal nature of the data has not been considered in contrast to the present study.

The absence of a significant association between PSNP participation and stunting in the older cohort unraveled by the present work is worth noting. When the PSNP was introduced, children in the younger cohort were about 4 years old while children in the older cohort were greater than 10 years old. By that age, the older children may be treated similar to adults and hence may lack the nutritional care accorded to younger children. According to Berhane et al [51], PSNP has no significant effect on the meal frequency for adults in contrast to its effect in underfive children. As also shown by Porter and Goyal[50], the impact of PSNP on nutritional status is higher for children exposed to the programme between 2 and 5 years of age.

Caution is required in the interpretation of the findings of the present analysis. As elaborated in the methods part, the Young Lives data was not designed to be nationally representative. Hence, cross-sectional prevalence, longitudinal prevalence and transition probabilities presented in this paper may not necessarily reflect the actual values of such measures in the population. They could be over- or under-estimates. They are meant primary to elucidate the longitudinal dynamics of stunting across the life course of children.

Besides, data on household food security status were available for the younger cohort since Round 3 and for the older cohort, only in Round 3. Additionally, data on interventions such as the Ethiopian Health Extension Programme (HEP) and provision of loans or credit to households are available only for the last two rounds (Rounds 4 and 5). Hence, the effects of food

security, the HEP and provision of loans/credits on stunting could not be investigated, nor could the effects of other determinants be adjusted for the effect of these variables. Further, data on infectious determinants of stunting such as malaria and intestinal parasitic infections were not available. Thus, the effects on stunting of these risk factors could not be determined, nor could the effects of other determinants be adjusted for the effects of infectious determinants. Consequently, while adjusting for wealth index may attenuate the confounding effect of some of these unmeasured factors, the results on determinants of stunting could still have been biased by failure to include the aforementioned factors. Further, the number of observations excluded from the analysis by case-wise deletion due to missingness on some of the analysis variables was considerable. This resulted in lower samples sizes in some analyses and might have introduced bias in some of the estimates. For example, some of the results on determinants of stunting might have been biased towards the null due to diminished power.

## Conclusion

The present analysis unravels that the cross-sectional prevalence of stunting fluctuates along the life course of Ethiopian children, but shows a general, though slight, decremental trend. Besides, the longitudinal prevalence reveals that the burden of stunting continues to be high in the life course of Ethiopian children. While children not stunted early in life have high probability of growing into non-stunted adults, children stunted early in life are at a higher risk of growing into stunted adults. Increase in the children's age decreases the risk of stunting to some extent. Male and rural children and children of illiterate mothers are at a higher risk of stunting. Children from households with illiterate heads also have higher risk of stunting. Younger children from socioeconomically disadvantaged households are at higher risk of stunting, but this doesn't seem to be the case for older children. Further, PSNP significantly reduces the risk of stunting in younger children, but not in older children. Interventions designed to prevent stunting and cut the transgenerational cycle of stunting should commence as early in the life of a child as possible, preferably since in-utero.

## Acknowledgments

Disclaimer: The views expressed here are those of the author(s). They are not necessarily those of, or endorsed by, Young Lives, the University of Oxford, DFID or other funders.

The data used in this publication come from Young Lives, a 15-year study of the changing nature of childhood poverty in Ethiopia, India (Andhra Pradesh and Telangana), Peru and Vietnam (www.younglives.org.uk). Young Lives has been core-funded by UK aid from the Department for International Development (DFID).

## Author Contributions

**Conceptualization:** Ayalew Astatkie.

**Data curation:** Ayalew Astatkie.

**Formal analysis:** Ayalew Astatkie.

**Investigation:** Ayalew Astatkie.

**Methodology:** Ayalew Astatkie.

**Project administration:** Ayalew Astatkie.

**Resources:** Ayalew Astatkie.

**Software:** Ayalew Astatkie.

**Validation:** Ayalew Astatkie.

**Visualization:** Ayalew Astatkie.

**Writing – original draft:** Ayalew Astatkie.

**Writing – review & editing:** Ayalew Astatkie.

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
