## [Decision Letter · Decision Letter 0]

3 Dec 2019

PONE-D-19-29316

Dynamics of stunting from childhood to youthhood in Ethiopia: Evidence from the Young Lives panel data (rounds 1-5)

PLOS ONE

Dear Dr. Ayalew Astatkie,

Thank you for submitting your manuscript to PLOS ONE. After careful consideration, we feel that it has merit but does not fully meet PLOS ONE’s publication criteria as it currently stands. Therefore, we invite you to submit a revised version of the manuscript that addresses the points raised during the review process.

I would like to applaud author for undertaking this outstanding work. The topic is timely and relevant to address issues related to stunting. As it has been clearly indicated, (Ethiopia) country profile on nutrition and child stunting trend, stunting is an indicator of the devastating result of malnutrition in early childhood. It is one of the subjects of public health importance in vast majority of developing countries like Ethiopia. That being said, I would like to provide opinions that would help you address some of the issues in this manuscript.

Abstract:

It is brief and precisely depicts the overall study 

Background:

It has already included literature that help to illustration the burden of malnutrition particularly in Ethiopia. Perhaps, it would be good if you add in more literature related to stunting to provide further insight about the subject, since stunting is one of the highly researched topic in Ethiopia and abroad. Otherwise, I found the literature in this section can expound the subject vividly.

Methods:

This section encompasses necessary study tools that help to conduct the research. I found it technically sound to provide a legit result.

Result:

The only gap I have seen in this section probably related to Socio-demographic description. You have tried to incorporate many textual details related to socio-demographic characteristics and the table inserted in between the paragraph, which has to be put in at the end, so that you can clearly show the analysis either way. On the other hand by minimizing the paragraph you can you can avoid repeating and too much details.  Otherwise, the analysis clear and technically sound.

Discussion:

Well written and included literature that help to make relevant argument. 

Conclusion:

It has been drawn based the finding. However, I wonder what recommendation the authors would provide from this work.

We would appreciate receiving your revised manuscript by Jan 17 2020 11:59PM. To enhance the reproducibility of your results, we recommend that if applicable you deposit your laboratory protocols in protocols.io, where a protocol can be assigned its own identifier (DOI) such that it can be cited independently in the future. For instructions see: http://journals.plos.org/plosone/s/submission-guidelines#loc-laboratory-protocols

We look forward to receiving your revised manuscript.

Kind regards,

Solomon Assefa Woreta

Academic Editor

PLOS ONE

Journal Requirements:

1. Please ensure you have thoroughly discussed any potential limitations of this study within the Discussion section.

Additional Editor Comments (if provided):

I would like to applaud author for undertaking this outstanding work. The topic is timely and relevant to address issues related to stunting. As it has been clearly indicated, (Ethiopia) country profile on nutrition and child stunting trend, stunting is an indicator of the devastating result of malnutrition in early childhood. It is one of the subjects of public health importance in vast majority of developing countries like Ethiopia. That being said, I would like to provide opinions that would help you address some of the issues in this manuscript.

Abstract:

It is brief and precisely depicts the overall study

Background:

It has already included literature that help to illustration the burden of malnutrition particularly in Ethiopia. Perhaps, it would be good if you add in more literature related to stunting to provide further insight about the subject, since stunting is one of the highly researched topic in Ethiopia and abroad. Otherwise, I found the literature in this section can expound the subject vividly.

Methods:

This section encompasses necessary study tools that help to conduct the research. I found it technically sound to provide a legit result.

Result:

The only gap I have seen in this section probably related to Socio-demographic description. You have tried to incorporate many textual details related to socio-demographic characteristics and the table inserted in between the paragraph, which has to be put in at the end, so that you can clearly show the analysis either way. On the other hand by minimizing the paragraph you can you can avoid repeating and too much details. Otherwise, the analysis clear and technically sound.

Discussion:

Well written and included literature that help to make relevant argument.

Conclusion:

It has been drawn based the finding. However, I wonder what recommendation the authors would provide from this work.

Reviewers' comments:

Reviewer's Responses to Questions

**Comments to the Author**

1. Is the manuscript technically sound, and do the data support the conclusions?

Reviewer #1: Yes

Reviewer #2: Yes

Reviewer #3: Yes

2. Has the statistical analysis been performed appropriately and rigorously? 

Reviewer #1: Yes

Reviewer #2: Yes

Reviewer #3: Yes

3. Have the authors made all data underlying the findings in their manuscript fully available?

Reviewer #1: Yes

Reviewer #2: Yes

Reviewer #3: Yes

4. Is the manuscript presented in an intelligible fashion and written in standard English?

Reviewer #1: Yes

Reviewer #2: Yes

Reviewer #3: Yes

5. Review Comments to the Author

Reviewer #1: Comments to the Author

Originally this paper was intended to address the dynamics of the stunting among childhood and youthhood in low income country where a child stunting is a public health problem. It provides a clue for program and policy makers to revisit the policy and programs designed to tackle child stunting in low income country. The paper is well written with adequate use of statistical analysis.

Relevance of the paper: It provides evidence how far stunting reduction programs are effectiveness and lessons within the context of PSNP program.

Methodological aspects:

• The methodological parts do not clearly depict the sources of data used in this specific paper. The author should consider specifying in study design as he had been analyzed the secondary data from……………………………….

• The author did not mention the specific woredas/districts where the original studies were done. For example the name of regions and then 20 districts from these regions were mentioned but it’s not clear which these 20 districts were. Thus the author should mention briefly the name and some sociodemographic and economic characteristics of the districts where the original studies were conducted.

• This specific paper mainly focused on studies done on childhood and youthhood. However, in paper there are inconsistency of the childhood and youthhood. For example, the author used a word ‘’children’’ or ‘’younger’’ and ‘’older’’ cohorts. I recommend to the author to make consistent throughout the whole document.

• It is obvious that the seasonal variation of data collection might have effect on anthropometric status if children. So that author should mention the time/seasonality of original studies data collection. Or the author can summarize in table of the ‘’1-5’’ studies by survey timing and round.

• In line 117, the author categorized stunting as ‘’ not stunted (HFA z-score of greater than or equal to -2), moderately stunted (HFA between -3 and -2), and severely stunted (a HFA of less than -3).’’. We cannot differentiate the moderately stunted child from severely stunted child. It is recommendable to classify stunting in to only two categories as ‘’Not stunted’’ or ‘’Normal’’ and ‘’Stunted.’’ Thus I recommend the author has to reconsider the classification. This might require redoing the analysis but I imagine it will not make significant changes to the results. But it is up to author’s decision which way is appropriate.

Results:

• The results sections were well written. However, I encourage the author to give some time to proofreading of the whole document including spacing between texts and citations and editing of all tables.

• The tables should be prepared based on the journal’s guideline.

• There should no lines between numbers and variables presented in the tables.

• Use single line spacing for all table titles

• To facilitate reading the data contained in the tables, use intermediate lines to separate the analysis variables and position the numbers on the right.

• There should be appropriate headings for each table. For example in 198, table 1: author mentioned only children as study participants but in table there is an older child too. So it has to be rephrased in order to make more inclusive of the study population.

• Table 1: is too long which covers almost four pages. It is difficult to easily understand the tables because it is too long and crowded. So that I recommend the author should give a time to make more clear and attractive as well as split long tables in to different tables.

• I suggest placing the titles next to both Figures-. Add caption/legend if necessary.

• Standardize the size and font used throughout the text

Reviewer #2: General Comment: Generally, I found that this paper is well organized and addressed one of the important public health problem in the country. The design and analyses employed in this study were appropriate and almost all the analyses were computed following the recommended procedures. In addition, the level of English language used in this document can be considered as adequate and might not need serious editorial revision at this level. Despite these; I have the following very few points which need clarification or justification;

Line 93: It says “Though the sample size was not incepted to be nationally representative…”. Also you described that a maximum of five districts from each selected regions and one kebele from each selected districts were included in the study. Having this non-representative sample, how can you justify that your study assessed dynamics of stunting in the Ethiopia? Don’t you think your title is a bit broader than the actual aspect addressed in the study? How do you justify your conclusion on line 475 about the whole of Ethiopian children while accepting the non-representativeness of the sample?

Line 95 and 96: Since it describe about EDHS and WNS you should include reference #3 with the indicated references (15,16)

Line 122: Change “– viz.” into “viz;”

Line 130: Your independent variables include age of the mother and age of the household head. Theses to variables have strong correlation because the age of a mother and age of her husband will have a positive relationship in the real world. In addition, what if the mother was also the head of the household; did you used two similar values as values of two independent variables? Have you considered the issue of multicolinearity especially with these independent variables? On line 163; you mentioned that father’s age was excluded. Was it different from age of the HH head?

Line 143: Change is to was “…data on which this article was based…”

Line 190: Remove “the” from “…the children involved…”

Line 191: Change “..are presented in Table 1” to “..were presented…”

Line 190-197: unnecessary capitalization of “Round”

Line 284: Inside the table the AOR for above grade 8 was 0.69 (0.47, 1.00). the 95%CI contains 1 inside but labeled as significant.

Line 284: The table should include only subjects with no missing data for all the included variables in the final model unlike the descriptive table. You also mentioned that variables with missing information were excluded.

Reviewer #3: Review comments

Dear Journal Editor,

Thank you for inviting me to review this manuscript: “Dynamics of stunting from childhood to youthhood in Ethiopia: Evidence from the Young Lives panel data (rounds 1-5)”

The paper addresses a pressing public health challenge in the study setting and of importance.

General comments

1. The author defined stunting and its global and national burden, trends, and determinants. However, upstream and downstream factors contributing to the steady decline in stunting over years were nor addressed in detail. It would give better understanding about the epidemiology of stunting for the readers of the manuscript if the author could include structural and other factors that have contributed to a disproportionate burden of stunting between regions within the country.

2. The author reported one of the interventions to ensure food security and combating stunting such as Productive Safety Net program; however, the time periods for the intervention was not mentioned. Were the study area, kebeles or study subjects beneficiaries of the intervention? Is there information about how long the households were included in the program? If so, the data acquisition process should be reported. In comparison of stunting among households with PSNP, did the author looked for the presence of other similar interventions in the area that could increase the effect size in reducing stunting.

Materials and methods

3. Who did collect the data? The author should have clearly described how the data were obtained. Who did a follow-up of the cohorts? Though the data sources were indicated in the given references, the author did not describe the cohort profile for the study subjects selected for the study (number/proportion of loss-to-follow up due to various reasons, deaths etc).

Discussion. The limitations of the study should be discussed in more detail.

6. PLOS authors have the option to publish the peer review history of their article (what does this mean?). If published, this will include your full peer review and any attached files.

Reviewer #1: No

Reviewer #2: Yes: Fanuel Belayneh Bekele

Reviewer #3: Yes: Mesay Hailu Dangisso

---

## [Author Response · Author response to Decision Letter 0]

3 Jan 2020

Journal requirements

Comment: Please ensure you have thoroughly discussed any potential limitations of this study within the Discussion section.

Response: Potential limitations of the study are discussed in the last two paragraphs of the manuscript. 

Comments from the editor

Comment: I would like to applaud author for undertaking this outstanding work. The topic is timely and relevant to address issues related to stunting. As it has been clearly indicated, (Ethiopia) country profile on nutrition and child stunting trend, stunting is an indicator of the devastating result of malnutrition in early childhood. It is one of the subjects of public health importance in vast majority of developing countries like Ethiopia. That being said, I would like to provide opinions that would help you address some of the issues in this manuscript.

Response: Thank you for the kind and encouraging words. Also thank you for the positive comments which are meant to improve the manuscript.

Comment: Abstract: It is brief and precisely depicts the overall study.

Response: Thank you for the positive assessment of the abstract.

Comment: Background: It has already included literature that help to illustration the burden of malnutrition particularly in Ethiopia. Perhaps, it would be good if you add in more literature related to stunting to provide further insight about the subject, since stunting is one of the highly researched topic in Ethiopia and abroad. Otherwise, I found the literature in this section can expound the subject vividly.

Response: Thank you for this comment. Now the arguments in the introduction are substantiated by citing additional relevant literature. Besides, literature related to infectious determinants of stunting has been included.

Comment: Methods: This section encompasses necessary study tools that help to conduct the research. I found it technically sound to provide a legit result.

Response: Thank you for the positive assessment of the methods of the manuscript.

Comment: Result: The only gap I have seen in this section probably related to Socio-demographic description. You have tried to incorporate many textual details related to socio-demographic characteristics and the table inserted in between the paragraph, which has to be put in at the end, so that you can clearly show the analysis either way. On the other hand by minimizing the paragraph you can you can avoid repeating and too much details. Otherwise, the analysis clear and technically sound.

Response: Thank you for this comment too. Some unnecessary textual descriptions are now removed from the sociodemography section. However, placement of tables within the manuscript is in accordance with PLoS ONE’s submission requirements, which states “Place each table in your manuscript file directly after the paragraph in which it is first cited (read order).” (https://journals.plos.org/plosone/s/submission-guidelines#loc-figures-and-tables).

Comment: Discussion: Well written and included literature that help to make relevant argument.

Response: Thank you for this positive assessment.

Comment: Conclusion: It has been drawn based the finding. However, I wonder what recommendation the authors would provide from this work.

Response: Thank you for the positive assessment of the conclusion. Considering that stunting early in life is associated with a higher probability of stunting in later life and vice versa, and considering that interventions such as PSNP have significant positive effects when introduced in earlier life (under five years) than when introduced in later life, the key recommendation, as stated in the conclusion, is to commence interventions designed to tackle stunting as early in life as possible.

Comments from Reviewer 1

Comment: Originally this paper was intended to address the dynamics of the stunting among childhood and youthhood in low income country where a child stunting is a public health problem. It provides a clue for program and policy makers to revisit the policy and programs designed to tackle child stunting in low income country. The paper is well written with adequate use of statistical analysis.

Response: Thank you for the kind and encouraging remarks.

Comment: Relevance of the paper: It provides evidence how far stunting reduction programs are effectiveness and lessons within the context of PSNP program.

Response: Thank you for this remark, too.

Methodological aspects:

Comment: The methodological parts do not clearly depict the sources of data used in this specific paper. The author should consider specifying in study design as he had been analyzed the secondary data from……………………………….

Response: Thank you for this concern. The source of the data has already been described in the “Methods and Materials” section under “Study design and population”. Accordingly, the data source is the Young Lives panel study which follows approximately 12,000 children over several years in four low and middle income countries since 2002. A reference has also been already provided (reference # 27). The DOI for accessing the data is also provided in the references section (reference #27) as http://doi.org/10.5255/UKDA-SN-7483-3. The design of the study has also been already described as “a longitudinal panel study”. Though the present work is based on an analysis of secondary data, that doesn’t change the design of the study which originally generated the data.

Comment: The author did not mention the specific woredas/districts where the original studies were done. For example the name of regions and then 20 districts from these regions were mentioned but it’s not clear which these 20 districts were. Thus the author should mention briefly the name and some sociodemographic and economic characteristics of the districts where the original studies were conducted.

Response: Thank you for the comment. For reason of protecting the privacy of the study participants, Young Lives does not provide the actual names of the study localities. However, a description of the study sites with anonymized names is already provided elsewhere (https://assets.publishing.service.gov.uk/media/5b9a93c4ed915d665412ca27/ETHIOPIA-SurveyDesign-Factsheet-Jan18_0.pdf). For readers who need more information, references have been cited already (references # 25 & 26).

Comment: This specific paper mainly focused on studies done on childhood and youthhood. However, in paper there are inconsistency of the childhood and youthhood. For example, the author used a word ‘’children’’ or ‘’younger’’ and ‘’older’’ cohorts. I recommend to the author to make consistent throughout the whole document.

Response: Thank you for this comment as well. Yes, the study involves a follow-up of the study participants from childhood to youthhood (1-15 years for the younger cohort and 8-22 years for the older cohort). For much part of the follow-up, the study participants were in the age category of “children” while they belonged to the category of “youth” in the last one or two rounds. As a result, it is inevitable to use terms such “child/children” somewhere and “youth” elsewhere. However, as per the comments provided, wherever possible throughout the manuscript all-encompassing terms such as “study participants” have been replaced instead of “children” and/or “youth”. 

Comment: It is obvious that the seasonal variation of data collection might have effect on anthropometric status if children. So that author should mention the time/seasonality of original studies data collection. Or the author can summarize in table of the ‘’1-5’’ studies by survey timing and round.

Response: Thank you for this important concern. In all rounds, data were collected at similar times – between October and December. Hence, seasonal variation could not be analyzed. However, even if data were collected in different seasons, stunting is unlikely to show seasonal variations since it is a chronic phenomenon and apparent changes in stature may not occur within a short period of time. A description of the years in which the follow-up data were collected is already provided under “Study design and population” and references have been cited for readers who need more information. The months in which data were collected have now been included under “Methods of data collection”.

Comment: In line 117, the author categorized stunting as ‘’ not stunted (HFA z-score of greater than or equal to -2), moderately stunted (HFA between -3 and -2), and severely stunted (a HFA of less than -3).’’. We cannot differentiate the moderately stunted child from severely stunted child. It is recommendable to classify stunting in to only two categories as ‘’Not stunted’’ or ‘’Normal’’ and ‘’Stunted.’’ Thus I recommend the author has to reconsider the classification. This might require redoing the analysis but I imagine it will not make significant changes to the results. But it is up to author’s decision which way is appropriate.

Response: Thank you for this concern. The Young Lives data classifies stunting as not stunted, moderately stunted and severely stunted, which is consistent with WHO’s z-score cut-offs for classifying moderate and severe undernutrition (WHO 1997 & https://www.who.int/nutrition/topics/moderate_malnutrition/en/). Further, using ordinal categorization of stunting would provide much richer information about stunting than the “yes/no” (stunted/not stunted) dichotomy. 

Results:

Comment: The results sections were well written. However, I encourage the author to give some time to proofreading of the whole document including spacing between texts and citations and editing of all tables.

Response: Thank you for this comment. The manuscript has been read and re-read and any necessary revisions have been done throughout.

Comment: • The tables should be prepared based on the journal’s guideline.

• There should no lines between numbers and variables presented in the tables.

• Use single line spacing for all table titles

• To facilitate reading the data contained in the tables, use intermediate lines to separate the analysis variables and position the numbers on the right.

Response: Thank you for the detailed comments regarding table preparation. The tables are prepared as per PLoS ONE’s table preparation guideline and further edited as per the comments. Now all table titles are single-spaced and all numbers within cells are right-aligned. Which lines should appear in a table and which lines should be hidden is an issue to be handled during the article production. PLoS ONE’s table preparation guideline states, “Tables must be cell-based in Microsoft Word or embedded with Microsoft Excel.” Accordingly, the tables are prepared consistent with PLoS ONE’s guideline provided at https://journals.plos.org/plosone/s/file?id=80c1/PLOSOne_formatting_sample_main_body.pdf. 

Comment: There should be appropriate headings for each table. For example in 198, table 1: author mentioned only children as study participants but in table there is an older child too. So it has to be rephrased in order to make more inclusive of the study population.

Response: Thank you for this concern. In the title of Table 1, now the word “children” is replaced with the more inclusive phrase “study participants”. The titles of the other tables have also been revised.

Comment: Table 1: is too long which covers almost four pages. It is difficult to easily understand the tables because it is too long and crowded. So that I recommend the author should give a time to make more clear and attractive as well as split long tables in to different tables.

Response: Thank you for this comment. Yes, Table 1 is a bit long and contains much information. That is due to the nature of the data presented in the table which is about the sociodemographic profile of the two cohorts across all follow up periods. Since the information provided is about sociodemography, splitting the table into more than one table would not be sensible. When the table is formatted as per the journal’s style during production, it will fit in about two pages. PLoS ONE allows tables that span more than one page. In its table preparation guideline, it states “Tables do not have strict width and height requirements. Do not split your table or otherwise try to make the table appear within the manuscript margins if it does not fit on one page…. In the PDF version of the published article, very wide tables may be printed sideways, and long tables may span more than one page.” (https://journals.plos.org/plosone/s/tables).

Comment: I suggest placing the titles next to both Figures-. Add caption/legend if necessary.

Response: Thank you for this concern. The figures and titles were submitted in accordance with PLoS ONE’s requirement which states “Figure captions must be inserted in the text of the manuscript, immediately following the paragraph in which the figure is first cited (read order). Do not include captions as part of the figure files themselves or submit them in a separate document.” (https://journals.plos.org/plosone/s/submission-guidelines). 

Comment: Standardize the size and font used throughout the text

Response: Thank you for this concern. The figures and titles were submitted in accordance with PLoS ONE’s requirement which states “Figure captions must be inserted in the text of the manuscript, immediately following the paragraph in which the figure is first cited (read order). Do not include captions as part of the figure files themselves or submit them in a separate document.” (https://journals.plos.org/plosone/s/submission-guidelines). 

Comment: Standardize the size and font used throughout the text.

Response: Thank you. A consistent formatting is applied throughout the manuscript. 

Comments from Reviewer 2

Comment: Generally, I found that this paper is well organized and addressed one of the important public health problem in the country. The design and analyses employed in this study were appropriate and almost all the analyses were computed following the recommended procedures. In addition, the level of English language used in this document can be considered as adequate and might not need serious editorial revision at this level. Despite these; I have the following very few points which need clarification or justification.

Response: Thank you for the kind and encouraging words.

Comment: Line 93: It says “Though the sample size was not incepted to be nationally representative…”. Also you described that a maximum of five districts from each selected regions and one kebele from each selected districts were included in the study. Having this non-representative sample, how can you justify that your study assessed dynamics of stunting in the Ethiopia? Don’t you think your title is a bit broader than the actual aspect addressed in the study? How do you justify your conclusion on line 475 about the whole of Ethiopian children while accepting the non-representativeness of the sample?

Response: Thank you for this legitimate concern. Yes, the Young Lives sample was not intended to be nationally representative. It was rather intended to be suitable to generate rich data that would enable the investigation of the longitudinal dynamics of child welfare variables. As such, estimates obtained from the data such as prevalence of stunting may not be nationally representative. As already elaborated in the discussion section (second paragraph from the last), caution is required in interpreting the results. On the other hand, as already elaborated under “Sample size and sampling”, while the Young Lives sample was selected much care and effort has been taken to ensure representation of children of various background characteristics such as poverty levels, urban/rural mix and food deficit status. Consequently, the Young Lives sample has been show to be comparable with other nationally representative samples such as the Demographic and Health Survey (DHS) sample and the Welfare Monitoring Survey (WMS) sample. Further, the sample was selected from a base population which covers 96% of the general population in Ethiopia. Therefore, the results may still reflect the situation in the entire country but recognizing the nature of the sample is useful for readers while interpreting the results of the present study. 

Comment: Line 95 and 96: Since it describe about EDHS and WNS you should include reference #3 with the indicated references (15,16)

Response: Thank you. Reference #3 is the mini-DHS 2019 report of Ethiopia. It is cited in order to show the current status of stunting in Ethiopia. On the other hand, references # 23 & 24 [in the revised manuscript references # 25 & 26] are Young Lives’ publications which describe the methodology of the Young Lives study. These references (23 & 24, in the revised version 25 & 26) show that the Young Lives sample is comparable with the DHS and WMS samples. They are cited in the present work to substantiate the argument about the comparability of the Young Lives sample with nationally representative samples. Hence, reference #3 does not go with references # 23 & 24 (now 25 & 26).

Comment: Line 122: Change “– viz.” into “viz;”

Response: Thank you for pointing this out. “viz.” is now replaced with “viz.,” which is consistent with the example usage on Cambridge Dictionary (https://dictionary.cambridge.org/dictionary/english/viz). “viz;” could not be taken as the example usages provided on the Cambridge and Oxford online dictionaries use either “viz.,” or “viz.”. 

Comment: Line 130: Your independent variables include age of the mother and age of the household head. Theses to variables have strong correlation because the age of a mother and age of her husband will have a positive relationship in the real world. In addition, what if the mother was also the head of the household; did you used two similar values as values of two independent variables? Have you considered the issue of multicolinearity especially with these independent variables? On line 163; you mentioned that father’s age was excluded. Was it different from age of the HH head?

Response: Thank you for the legitimate concern. Yes, some of the variables seem to be correlated. Age of the mother and age of the household head correlated with a correlation coefficient (r) of 0.55. Similarly, age of the father and age of the household head were highly correlated with an r of 0.79. In the study’s setting (Ethiopia) husbands are generally regarded as household heads and female are considered heads in the absence of husbands. Hence, mother’s age is unlikely to be completely redundant with age of the household head. Hence, the two are expected to be correlated since as the husband (who is more likely the household head) gets older the wife (mother of the child) also gets older. Considering the correlated nature of the variables, the model was re-fit removing alternatively the mother’s age and the age of the household head. However, removal of either of the independent variables resulted in diminished model goodness-of-fit as evidenced by likelihood ratio test, Akaike information criterion (AIC) and Bayesian information criterion (BIC). As a result both variables were retained in the model. On the other hand, father’s age correlated highly with the age of the household head (r=0.79). Besides, there were a lot of missing observations for the variable “father’s age”. Hence, father’s age was removed from the multivariable model because of its redundancy with age of the household head and because of loss of thousands of observations when it is included in the analysis. 

Comment: Line 143: Change is to was “…data on which this article was based…”

Response: Thank you for the concern. The description in the mentioned sentence is about the present work. Hence, description in the present tense is more appropriate.

Comment: Line 190: Remove “the” from “…the children involved…”

Response: Thank you for this comment. From the preceding descriptions in the manuscript, it is clear for readers that the study participants are children. Hence, the use of the definite article “the” at the mentioned place is appropriate.

Comment: Line 191: Change “..are presented in Table 1” to “..were presented…”

Response: Thank you for the suggestion. The description in the mentioned sentence is about results presented in the present manuscript. Thus, the present tense is more appropriate than the past tense.

Comment: Line 190-197: unnecessary capitalization of “Round”

Response: Thank you for the concern. Round 1, Round 2, etc serve as proper nouns for each wave of data collection. Hence, the capitalizations are proper.

Comment: Line 284: Inside the table the AOR for above grade 8 was 0.69 (0.47, 1.00). the 95%CI contains 1 inside but labeled as significant.

Response: Thank you for pointing this out. The mentioned confidence interval embraced 1 when the AOR is rounded down to two decimal places. In 3 decimal places, the upper limit of the AOR is 0.999 and doesn’t embrace 1. Now it has been edited accordingly.

Comment: Line 284: The table should include only subjects with no missing data for all the included variables in the final model unlike the descriptive table. You also mentioned that variables with missing information were excluded.

Response: Thank you for this comment. The existence of missing observations in datasets is mostly the rule rather than the exception. In the present analysis, most of the variables have some missing observations. The missing observations are “true missing”; i.e., they were not generated by design such as “skip patterns” in the data collection questionnaire. As such, limiting the analysis only to variables with no missing observations is almost impossible. Case-wise deletion of missing observations was applied. However, variables with a lot of missing observations have been excluded from the analysis as already described in the methods section of the manuscript. Still, the cumulative number of observations excluded from the analysis by case-wise deletion due to missingness on some of the analysis variables is not ignorable. The implication of this situation has been discussed as a limitation in the last paragraph of the discussion section. 

Comments from Reviewer 3

Comment: Dear Journal Editor, 

Thank you for inviting me to review this manuscript: “Dynamics of stunting from childhood to youthhood in Ethiopia: Evidence from the Young Lives panel data (rounds 1-5)”. The paper addresses a pressing public health challenge in the study setting and of importance.

Response: Thank you for the positive assessment of the manuscript.

Comment: 1. The author defined stunting and its global and national burden, trends, and determinants. However, upstream and downstream factors contributing to the steady decline in stunting over years were not addressed in detail. It would give better understanding about the epidemiology of stunting for the readers of the manuscript if the author could include structural and other factors that have contributed to a disproportionate burden of stunting between regions within the country.

Response: Thank you for this comment. Possible factors which might explain the decline of stunting overtime in Ethiopia are addressed in detail in the discussion section (second paragraph). Factors which might account for the geographically disproportionate burden of stunting are described in the introduction of the manuscript (third paragraph). 

Comment: 2. The author reported one of the interventions to ensure food security and combating stunting such as Productive Safety Net program; however, the time periods for the intervention was not mentioned. Were the study area, kebeles or study subjects beneficiaries of the intervention? Is there information about how long the households were included in the program? If so, the data acquisition process should be reported. In comparison of stunting among households with PSNP, did the author looked for the presence of other similar interventions in the area that could increase the effect size in reducing stunting.

Response: Thank you for the legitimate concern. As already stated in the manuscript, the Productive Safety Net Programme (PSNP) was introduced in Ethiopia in 2005 (while, the Young Lives study was already under way – 3 years after Round 1 and about 1 year before Round 2). Once the PSNP was introduced, households were enrolled as beneficiaries of the PSNP in accordance with the beneficiary recruitment criteria. Accordingly, there were beneficiaries of the PSNP in the study localities as well, since recruitment of the study participants ensured representation of households with different food deficit status. As already described in the manuscript, data regarding participation in PSNP of households from which the study participants were recruited were collected as of round 3. The PSNP participation status was recorded as yes/no. There are no data regarding how long households have participated in PSNP. Hence, duration of participation in PSNP could not be addressed in the present analysis. The constructed Young Lives dataset also contains data about other interventions such as the Ethiopian Health Extension Programme and provision of loans or credit to households. However, such data are available only for the last two rounds (Rounds 4 & 5) and hence cannot be included in the analysis. Therefore, the effect on stunting of such factors could not be determined, nor could the effect of other determinants be adjusted for the effects of such factors. Now this fact has been discussed as a limitation in the last paragraph of the discussion section. 

Materials and methods

Comment: 3. Who did collect the data? The author should have clearly described how the data were obtained. Who did a follow-up of the cohorts? Though the data sources were indicated in the given references, the author did not describe the cohort profile for the study subjects selected for the study (number/proportion of loss-to-follow up due to various reasons, deaths etc).

Response: Thank you for this comment. Now additional description of who collected the data and how has been included in the second paragraph of the “Methods of data collection” section. However, who specifically the data collectors were in terms of academic qualification, profession, etc is not available from the Young Lives data documentation. The cohort profile is already provided in Fig 1.

Comment: Discussion. The limitations of the study should be discussed in more detail.

Response: Thank you for this comment. Now the limitations of the study have been discussed in more detail in the last two paragraphs of the discussion section.

---

## [Editor Report · Decision Letter 1]

29 Jan 2020

Dynamics of stunting from childhood to youthhood in Ethiopia: Evidence from the Young Lives panel data

PONE-D-19-29316R1

Dear Dr. Ayalew Astatkie,

We are pleased to inform you that your manuscript has been judged scientifically suitable for publication and will be formally accepted for publication once it complies with all outstanding technical requirements.

With kind regards,

Solomon Assefa Woreta

Academic Editor

PLOS ONE
---

## [Editor Report · Acceptance letter]

31 Jan 2020

PONE-D-19-29316R1 

Dynamics of stunting from childhood to youthhood in Ethiopia: Evidence from the Young Lives panel data 

Dear Dr. Astatkie:

I am pleased to inform you that your manuscript has been deemed suitable for publication in PLOS ONE. Congratulations! Your manuscript is now with our production department. 

With kind regards,

on behalf of

Dr. Solomon Assefa Woreta 

Academic Editor

PLOS ONE